# AI Control for Pasteurized Soft-Boiled Eggs

**DOI:** 10.3390/foods14183171

**Published:** 2025-09-11

**Authors:** Primož Podržaj, Dominik Kozjek, Gašper Škulj, Tomaž Požrl, Marjan Jenko

**Affiliations:** Laboratory for Mechatronics, Production Systems and Automation (LAMPA), Faculty of Mechanical Engineering, University of Ljubljana, 1000 Ljubljana, Slovenia; primoz.podrzaj@fs.uni-lj.si (P.P.); tomaz.pozrl@fs.uni-lj.si (T.P.)

**Keywords:** AI temperature control, artificial intelligence, machine learning, pasteurized soft-boiled eggs, robust reinforcement learning

## Abstract

This paper presents a novel approach to thermal process control in the food industry, specifically targeting the pasteurization and cooking of soft-boiled eggs. The unique challenge of this process lies in the precise temperature control required, as pasteurization and cooking must occur within a narrow temperature range. Traditional control methods, such as fuzzy logic controllers, have proven insufficient due to their limitations in handling varying loads and environmental conditions. To address these challenges, we propose the integration of robust reinforcement learning (RL) techniques, particularly the utilization of the Deep Q-Network (DQN) algorithm. Our approach involves training an RL agent in a simulated environment to manage the thermal process with high accuracy. The RL-based system adapts to different heat capacities, initial conditions, and environmental variations, demonstrating superior performance over traditional methods. Experimental results indicate that the RL-based controller significantly improves temperature regulation accuracy, ensuring consistent pasteurization and cooking quality. This study opens new avenues for the application of artificial intelligence in industrial food processing, highlighting the potential for RL algorithms to enhance process control and efficiency.

## 1. Introduction

### 1.1. AI Foods Processing

The growing integration of artificial intelligence (AI) across the food processing chain reflects a global trend toward intelligent, data-driven food processing systems. Recent research highlights AI applications in process automation, quality assurance, energy optimization, supply chain management, and food safety, spanning sectors such as meat processing, fermentation, and heat drying. This evolution, and technical background are well documented in a broad range of recent reviews [1,2,3,4,5,6,7].

In food processing, key concerns have become sustainability, safety, and quality. AI technologies contribute significantly to reducing resource consumption, minimizing waste, and improving overall process efficiency [8].

AI-based control systems at minimum support real-time process regulation, adaptive decision-making, and energy and material use optimization. These systems are applied in diverse contexts [9], including fermentation control, meat production, drying operations, and thermal processing, often outperforming traditional rule-based controllers [10].

In the context of food thermal processing, control over temperature profiles is essential to ensure both microbial safety and desired texture. Research in phase-change heat transfer demonstrates how AI models can capture complex dynamics of heating and cooling to support optimized control strategies [11]. Furthermore, digital twin frameworks have been proposed for thermal food processes, leveraging reduced-order models to implement autonomous and real-time control with minimal computational overhead [12]. These examples underscore the broader applicability of AI in managing thermal treatment regimes across food applications.

The current study represents, to the best of the authors’ knowledge, the first successful implementation and proof-of-concept of reinforcement learning for precise temperature control, where the agent was trained exclusively in a simulated environment using domain randomization and then applied directly to the real system without additional training or fine-tuning. In the real environment, the agent achieved sub-0.1 °C accuracy. The main novelties of this work include: (1) the use of wide parameter ranges in domain randomization, resulting in a highly robust agent model; (2) the ability to cope with relatively large response delays between control actions and temperature feedback; and (3) achieving sub-0.1 °C precision within a 2 min period in controlling the temperature of a relatively large liquid volume (>7 L of water) using only a fast-responsive on/off-type controller. This work is therefore among the rare successful cases where a reinforcement learning agent trained and tuned solely in simulation, demonstrated consistent performance when transferred to a real food-processing environment.

### 1.2. Pasteurized Soft-Boiled Eggs

The key factor in preparing eggs for consumption is increased temperature, with boiling, steaming, baking, frying, and poaching being the most common ways of preparing eggs [13,14]. In addition to changing the molecular structure of the eggs, the preparation process must kill the bacteria potentially present in the eggs. Specifically, salmonella is the bacteria causing the most problems. When food sources of salmonella infections have been studied, the consumption of raw or undercooked eggs has repeatedly been reported as a major factor related to both sporadic cases and outbreaks [15,16]. The range of appropriate temperatures and times for pasteurized soft-boiled egg preparation have been studied for quite some time [17,18,19,20,21].

The current state-of-the-art control algorithm for temperature regulation in pasteurized soft-boiled egg cooking was previously developed by two of the authors of this paper [22]. A thorough search for improved approaches revealed no significant advances beyond this earlier work.

Fuzzy logic was originally introduced to mimic human reasoning and has been widely applied to processes in which human operators traditionally perform effectively. Cooking represents a clear example of such a skill, which motivated the use of fuzzy logic in our original study. In that work, the controller was designed using intuitive rules, such as “If the temperature is too low, increase the heater power”, and it performed reliably in controlled laboratory conditions.

A human operator, however, continuously adapts control decisions to the prevailing circumstances, drawing on years of experience and real-time observation of the apparatus and the process. In principle, a process controller could also incorporate not only process variables but also environmental variables in real time, yet this would greatly increase system complexity and require extensive scenario planning in advance.

Our proposed approach addresses this limitation by introducing probability densities for environmental variables across wide ranges. The process controller is therefore designed to operate optimally under arbitrary combinations of environmental conditions within these probability distributions. Importantly, this does not increase system complexity (electronics, mechanics, sensors, HMI), although the performance becomes highly robust, with the process being almost insensitive to environmental fluctuations. The resulting signal-to-noise ratio approaches the ideal case where disturbances are effectively decoupled from the thermal process. This represents a novelty in precise thermal control for food processing, developed and demonstrated in this study for the case of pasteurized soft-boiled eggs.

In our approach, a reinforcement learning (RL) agent is trained in an environment defined by parameter probability distributions, enabling it to optimally govern the thermal process. The agent experiences a large number of cooking episodes with different parameter values (within their probability ranges), during which it autonomously develops its control strategy rather than relying on manually defined rules. This makes RL-based control inherently robust to environmental fluctuations. The potential of RL has already been demonstrated in other complex domains, such as games like Chess and Go, where agents have achieved superhuman performance without the explicit programming of rules. Similarly, in the cooking process studied here, RL delivers robust control without predefined, human-crafted rules, relying instead on strategies that emerge during training.

From our previous work the curvature presented in Figure 1 is a reference curve [23,24]. In this figure, the duration of period A is influenced by the batch thermal load, period B lasts 1 min, period C lasts 30 s, period D lasts 10 min, period E lasts 5 s, and period F may last up to 6 h, as soft-boiled eggs are intended to be kept warm until consumption.

We verified Figure 1’s soft-boiled eggs temperature profile for proper bacteriological [25,26], rheological [27,28] and organoleptic properties.

The schematic of our egg cooker is shown in Figure 2.

Eggs are placed in hot water heated by a heating coil. The maximum time derivative of the temperature is therefore determined by the power of the heating coil. To make the temperature in the hot water reservoir as uniform as possible, a hot water pump is added to circulate the hot water.

In theory, the temperature can be decreased simply by stopping the heating; then, due to the heat losses, the temperature of the hot water will start to decrease. This process is, however, too slow. Therefore, we added a special reservoir with cold water surrounding the hot water. The electromagnetic valve is used to mix the hot water with the cold water. The longer the valve is open, the more cold water is added. In this manner, we can cool the hot water much faster. The cooling effect, however, depends on the temperature of the hot water, as demonstrated in Figure 3, left. Cooling from 90 °C and from 60 °C can be distinguished as follows: the time delay in period A differs from that in period B, and the temperature slope in period C differs from that in period D.

The most common feedback control systems typically employ a PID controller. In this case, the actuating action is assumed to be symmetrical. With our system, this is not the case, as seen in Figure 3, right. Heating and cooling from 60 °C can be distinguished as follows: the time delay in period E differs from that in period F, and the temperature slope in period G differs from that in period H. The actuating intensity (the temperature derivative) is very different for both cases (heating and cooling). An additional problem is posed by very different time delays for both cases. So, it is clear that a common PID controller cannot be used [29]. In our previous study, a fuzzy controller was used to solve the above-mentioned issue [22]. The resulting product is shown in Figure 4.

While the product proved successful, intensive real-world use revealed limitations:Load sensitivity: The control algorithm performs well under full-load conditions (maximum number of eggs and water). However, when operating with partial loads—frequently required in practical use—the reduced heat capacity alters system dynamics, resulting in suboptimal control performance.Ambient temperature influence: Elevated room temperature, especially after prolonged operation (eggs can stay warm in the apparatus up to 6 h), contributes to firmer egg texture.Heater fouling: Limescale accumulation on the heater surface degrades heat transfer efficiency, thereby altering the thermal response of the system and impairing the accuracy of temperature control.

Based on these findings, and artificial intelligence development, we developed a controller based on artificial intelligence, aimed at overcoming the above-mentioned shortcomings of the current system.

## 2. Reinforcement Learning Based Temperature Control

### 2.1. Etymology and Conceptual Origins of the Reinforcement Learning

The term reinforcement learning (RL) originated at the intersection of behavioral psychology and computational learning theory.

RL is derived from the verb “to reinforce”, meaning to strengthen or support. Its Latin roots—*re* (again) and *infortis* (strong)—suggest the idea of making something stronger through repeated application. This concept was first formalized in the late 19th century [30] (pp. 87–103). In shaping the behavior is emphasized the role of consequences—rewards and punishments.

In artificial intelligence, the term RL gained prominence in the 1980s and 1990s, largely through the seminal work of the authors cited in [31] and [32] (pp. 13–22). Their formulation of RL as a computational framework mirrors the psychological model: an agent receives rewards from the environment and adapts its policy to maximize cumulative reward over time. Unlike supervised learning, which relies on labeled data, or unsupervised learning, which infers structure from input data, RL emphasizes trial-and-error interaction with delayed and often sparse feedback.

### 2.2. RL Components

RL is conducted in a sequence of actions that (a) should cause a good outcome immediately after taking each action and (b) have a good integral outcome after a series of actions.

The following are essential RL components [33] (pp. 34–37), [34] (pp. 1458–1557):The environment.The agent interacting with the environment.Observation—Before the agent takes an action, it observes the environment state, which serves as a part of information, based on which the agent decides which action to take.Action—An agent is interacting with the environment by taking actions based on its observations. Actions taken by the agent typically affect the state of the environment.Step—One step in the reinforcement system is defined as a cycle consisting of (a) the agent receiving observational information (and a reward, in the learning phase), (b) deciding which action to take, (c) taking the action, and (d) updating the environment state based on the agent’s action.Episode—One episode consists of multiple steps. For example, in a chess board game, one move on the chessboard is considered one step, and one game of chess is considered one episode.Reward—In the learning phase, i.e., the training phase, the agent is receiving rewards. These rewards serve as feedback for the agent, based on which the agent knows how well it is acting. The agent aims to collect as high a cumulative reward as possible throughout the steps of an episode. The agent learns to take actions that will lead to collecting the highest cumulative reward by the end of the episode.Policy—In the learning phase, the agent interacts with its environment with the goal of learning behavior policy that will earn him a high cumulative reward at the end of each episode. The agent is taking steps, one after another, episode after episode. The policy, which the agent is developing and refining throughout a sequence of episodes, defines a decision system. Based on current and previous observations, the decision system defines what action needs to be taken at each step: inputs are current and previous observations; output is the action to be taken. The decision system contains many learnable parameters, the values of which are defined during the learning phase.

Different RL algorithms have different optimization algorithms and different capacities for implementation of the decision-system equations. Not every RL algorithm can be used for every RL case, but typically, more than one RL algorithm can be used for a specific RL case. We selected the discrete-action Deep Q Network agent [35,36] (DQN, where Q(s,a) is a function that estimates the expected cumulative reward for taking action in state s), in the form of a feedforward Multilayer Perceptron (MLP) [37] optimized for low-dimensional (non-image) inputs. The agent’s network consists of an input layer, two hidden layers with 64 neurons each, a ReLU (Rectified Linear Unit) activation function, and an output layer. This network is fully connected, in which each neuron in one layer is connected to every neuron in the next layer; this is the standard setup for MLPs.

### 2.3. The Agent and the Egg Cooker

In the egg cooking application, the agent’s environment is the cooking process. Its actuators are a 3.6 kW heater and an on/off valve for cooling eggs, immersed in heated water, with an inrush of cold water, as shown in Figure 2. To ensure a satisfactory degree of flexibility in controlling the thermal process, we decided upon four actions that the agent chooses from in each step:Heater off, valve closed;Heater on, valve closed;Heater off, valve open;Heater on, valve open.

Actions 1, 2, and 3 correspond to the most intuitive control strategies: no action, heating, and cooling. The rationale for including action 4—attenuated cooling—is based on the observation that cooling (enabled by valve opening) produces a significantly greater effect on the medium temperature change than heating.

During real-system testing of the trained RL agent, we found that cooling was excessively strong for precise temperature regulation, even though action 4 was available in the agent’s action space. To address this, we introduced a minor adjustment during deployment: when the absolute temperature error was below 2 °C (i.e., $\left|T_{i}^{m}-T_{i}^{ref}\right|<2\;{}^{\circ}C$), the cooling valve was restricted to 20% open, rather than 100%, to reduce thermal undershoot.

The time of a single step, i.e., the time difference ∆*t* between two consecutive steps, is 1 s. Given the relatively slow dynamics of cooking, prior experience indicates that a decision period of 1 s is adequate.

One episode consists of maximum 5000 steps (5000 s, i.e., 83.33 min). The episode ends sooner if the target temperature profile is reached (heating up to 90 °C, then 1 min at 90 ± 2 °C, 10 min at 60 ± 2 °C, and 10 min at 57 ± 2 °C). The duration of the first phase—heating up to 90 °C, depends on the heat capacity and heating power in the episode.

In the 5000 steps, the target temperature profile needs to be achieved as accurately as possible. Observation *o*, given to the agent from the environment in each step, is defined in Equation (1):(1)$$\begin{array}{c} o=\left[T_{i}^{m},T_{i}^{ref},T_{i}^{m}-T_{i}^{ref},{slope}_{i},{heating}_{i},{cooling}_{i},\right.\\ T_{i-1}^{m},T_{i-1}^{ref},T_{i-1}^{m}-T_{i-1}^{ref},{slope}_{i-1},{heating}_{i-1},{cooling}_{i-1},\\ T_{i-2}^{m},T_{i-2}^{ref},T_{i-2}^{m}-T_{i-2}^{ref},{slope}_{i-2},{heating}_{i-2},{cooling}_{i-2},\\ T_{i-3}^{m},T_{i-3}^{ref},T_{i-3}^{m}-T_{i-3}^{ref},{slope}_{i-3},{heating}_{i-3},{cooling}_{i-3},\\ \ldots \\ \left.T_{i-9}^{m},T_{i-9}^{ref},T_{i-9}^{m}-T_{i-9}^{ref},{slope}_{i-9},{heating}_{i-9},{cooling}_{i-9}\right] \end{array}$$
where $T_{i}^{m}$ is the measured temperature at the *i*th step, $T_{i}^{ref}$ is the reference temperature at the *i*th step, ${slope}_{i}$ is the temperature increasing/decreasing rate at the *i*th step calculated for the last 10 steps (last 10 s), and ${heating}_{i}$ and ${cooling}_{i}$ denote the heating and cooling states (on, off), respectively, at the *i*th step.

The observation function in DQN learning is defined intuitively, based on personal judgment and prior experience. The content and scope of the observation must be optimally balanced. An overly extensive observation introduces noise and does not aid learning, whereas an overly limited observation hinders the agent’s ability to learn effectively. Successful learning is not guaranteed for any given choice of observation function. If learning progress is poor or even diverges, it may be necessary to reconsider and modify the observation function. However, in thermal systems, key data include temperature, the slope of the temperature profile, and the temperature response delay. The observation function should therefore meaningfully comprise these quantities.

At each step, together with the above-defined observation *o*, the agent receives a reward *r*, as shown in Equation (2):(2)$$r=-1-\left|T_{i}^{m}-T_{i}^{ref}\right|+\left(\left|T_{i-1}^{m}-T_{i-1}^{ref}\right|-\left|T_{i}^{m}-T_{i}^{ref}\right|\right)+\sum_{i=1}^{5}b_{i}+p_{1}+\sum_{i=1}^{3}s_{i}$$
where $b_{1}$, $b_{2}$, $b_{3}$, $b_{4}$, $p_{1}$, $s_{1}$, $s_{2}$, and $s_{3}$ are bonuses or punishments defined as:$$b_{1}=\left\{\begin{array}{c} 10;\;\left|T_{i}^{m}-T_{i}^{ref}\right|<1\;{}^{\circ}C\\ 0;\;\left|T_{i}^{m}-T_{i}^{ref}\right|\geq 1\;{}^{\circ}C \end{array}\right.\;{\;\;\;\;\;\;b}_{2}=\left\{\begin{array}{c} 100;\;\left|T_{i}^{m}-T_{i}^{ref}\right|<0.5\;{}^{\circ}C\\ 0;\;\left|T_{i}^{m}-T_{i}^{ref}\right|\geq 0.5\;{}^{\circ}C \end{array}\right.\;{\;\;\;\;\;\;b}_{3}=\left\{\begin{array}{c} 150;\;\left|T_{i}^{m}-T_{i}^{ref}\right|<0.1\;{}^{\circ}C\\ 0;\;\left|T_{i}^{m}-T_{i}^{ref}\right|\geq 0.1\;{}^{\circ}C \end{array}\right.\;{\;\;\;\;\;\;\;\;\;b}_{4}=\left\{\begin{array}{c} 200;\;\left|T_{i}^{m}-T_{i}^{ref}\right|<0.05\;{}^{\circ}C\\ 0;\;\left|T_{i}^{m}-T_{i}^{ref}\right|\geq 0.05\;{}^{\circ}C \end{array}\right.\;{\;\;\;\;\;\;\;\;b}_{5}=\left\{\begin{array}{c} 100;\;\left|T_{i}^{m}-T_{i}^{ref}\right|<0.5\;{}^{\circ}C\;⋀\;\left|T_{i}^{m}-T_{i-1}^{m}\right|<0.05\;{}^{\circ}C\\ 0;\;other \end{array}\right.\;p_{1}=\left\{\begin{array}{c} 0;\;if\;{heat}_{i}={heat}_{-i}⋀{cooling}_{i}={cooling}_{-i}\\ -10;\;if\;{heat}_{i}\neq {heat}_{-i}⋁{cooling}_{i}\neq {cooling}_{-i} \end{array}\right.\;{\;\;\;\;\;\;\;\;s}_{1}=\left\{\begin{array}{c} -100;\;\left(T_{i-1}^{m}-T_{i-1}^{ref}\right)<0⋀{cooling}_{i}==1\\ 0;\;other \end{array}\right.\;{\;\;\;\;\;\;\;\;s}_{2}=\left\{\begin{array}{c} -100;\;\left(T_{i-1}^{m}-T_{i-1}^{ref}\right)>0.1\;{}^{\circ}C⋀{heat}_{i}==1\\ 0;\;other \end{array}\right.\;{\;\;\;\;\;\;\;\;s}_{3}=\left\{\begin{array}{c} 10;\;{heat}_{i}==0\;⋀{cooling}_{i}==0\\ 0;\;other \end{array}\right.$$

For the specific task of precise temperature control profile—featuring both heating and cooling phases as well as temperature response delays due to heat transfer—key components of the reward include the proximity of the actual temperature to the reference temperature and the prudence of control actions. The agent is rewarded for applying control actions gently and reducing the temperature error.

The reward function, defined in Equation (2), was designed based on process-specific knowledge, including the thermal dynamics of the system, pasteurization safety requirements, and the influence of temperature trajectories on egg texture. It was then fine-tuned through simulation experiments. Once established, it was fixed and applied directly to the real system without any additional adjustment. In this way, the simulation served not only as the training environment for the RL agent but also as a testbed for reward function design, which subsequently performed well on the physical system without further tuning.

The rationale of the “−1” term in the reward function (see the first term in Equation (2)) is to impose a small step penalty at each time interval. This discourages the agent from idling and ensures that it remains incentivized to act effectively throughout the cooking process.

The second term $\left|T_{i}^{m}-T_{i}^{ref}\right|$ guides the agent to lower the difference between the actual/measured and reference temperature.

The term $\left|T_{i-1}^{m}-T_{i-1}^{ref}\right|-\left|T_{i}^{m}-T_{i}^{ref}\right|$ provides a reward when the agent reduces the difference between the measured and the reference temperature from the previous to the current step.

The term $\sum_{i=1}^{5}b_{i}$, with relatively large reward values, promotes very precise tracking of the reference temperature profile.

The term $p_{1}$ is included to reduce switching between heating and cooling modes, since frequent switching can destabilize the process; therefore, minimizing the number of such transitions is desirable.

The last term $\sum_{i=1}^{3}s_{i}$ prevents the agent from taking inappropriate actions, such as activating cooling when the measured temperature is below the reference or heating when it is above. It also promotes operation in the most stable mode, in which neither heating nor cooling is activated.

When DQN agent training shows poor convergence, the first step is to adjust the reward function [38]. If this does not improve learning, the next step is to increase the amount of information included in the observation. This is followed by HIL (Hardware-In-the-Loop) testing, i.e., by controlling the physical system. If the agent performs poorly at this stage, the virtual model used during training may need to be improved. However, it is also possible—as was the case in our study—that the physical apparatus itself requires modifications, due to its’ previously unrecognized characteristics that must be improved.

To avoid policy learning in the process regions for which clear control rules are known, i.e., where the actual temperature is far away from the reference temperature, the RL agent controls the system only when the difference between the actual and reference temperature is less than 3 °C.

### 2.4. Learning in a Simulated Environment

A key limitation of reinforcement learning is the need for agents to undergo millions of interaction steps during training episodes to achieve acceptable performance. In many cases, it is not economical or even feasible to perform learning on the target system, since the learning process would be prohibitively time-consuming. The obvious solution to this problem is to conduct the learning phase in a simulated environment and then transfer the learned agent to the target system. The caveats of this solution are that the simulation needs to be at least a magnitude faster than the physical process, and the discrepancy between the process and its simulation must not critically influence the validity of the simulation results.

Also in this study, as will be demonstrated later with the RL results, it was not feasible to conduct learning on the target physical system. For this reason, the agent learned in the simulated environment described in this section.

The cooking process influencing parameters with variable values are:
Initial temperature of the external (cold) water, with constant mass;Initial temperature of the device casing;Ambient temperature;Initial temperature of the eggs;Egg size;Egg age;Number of eggs;Mass of the heated water;Initial temperature of the heated water;Electrical supply voltage ($P_{heater}=U^{2}/R_{heater}$), including variations from 230 V ± 15% (i.e., 195 V to 253 V, per EN 50,160 and IEC 61010-1 standards), and the voltage drop in a potential extension cord;Limescale deposits on the heating element.

Parameters 1, 2, and 3 are accounted for through a wide range of cold-water temperatures, as shown in Table 1.

Parameter 4, the initial egg temperature, is standardized at 4 °C (tolerance −0 °C to +1 °C) in industrial kitchens.

Parameter 5, egg size (S, M, L), is taken into account by adjusting the duration of temperature holding at 90 °C. In the egg cooker, this can be set through the user interface. In industrial kitchens, eggs are typically purchased in uniform size classes, and all eggs cooked simultaneously belong to the same size standard.

Parameter 6 is egg age. Even under optimal refrigeration conditions at 4 °C, chicken eggs undergo gradual physicochemical changes. A measurable increase in air cell size typically begins within 7 to 10 days post-laying, reflecting moisture and CO_2_ loss through the shell. These changes continue progressively, with observable deterioration of internal structures occurring beyond 3–4 weeks. In industrial kitchens, eggs are typically used within 10 days of laying.

Parameter 7, the quantity of eggs (1–30), is considered using the ratio of the specific heat capacities of the eggs and the water, within an extended range of heated water mass. The specific heat of water ($c_{p,\;water}$) is a fixed parameter with a value of 4.18 J/(kg K). The eggs mass is scaled into the water mass using a specific heat ratio (hen’s egg, $c_{p,\;egg}=\;3.30–3.70\;kJ/kgK$).

Parameter 9, the initial temperature of the heated water, does not influence the control process. Heating always begins with the maximum power, and the DQN-based heating control is activated once the temperature reaches 87 °C.

Parameter 10, the variation in electrical supply voltage, is accounted for by modeling the range of heater power.

Parameter 11, the presence of limescale on the heater, is considered through variations in the temperature lag of the heated water.

At the beginning of each episode, the domain randomization technique is used to set the values of the simulation model parameters, i.e., the mass of the heated water in the apparatus ($m_{water}$), heating power ($P_{heater}$), temperature of the cooling water ($T_{cold\;water}$), mass flow rate of the cooling water from the cooling system ($\dot{m}$), heating system temperature response delay (${delay}_{heating}$), and cooling system temperature response delay (${delay}_{cooling}$). This enables us to handle the issue of not knowing the exact values of these parameters in the real system and to make the control algorithm most robust to different combinations of initial conditions. At the beginning of each episode, the variable parameters values of the simulation model are randomly set within apparatus operational limits, as shown in Table 1. Temperature measurement noise was simulated with the Gaussian distribution around the actual temperature value.

A simplified linearized simulation model of the apparatus calculates the change in the temperature in each step using Equation (3),(3)$$T_{i}=T_{i-1}+\left(∆T_{heating}+∆T_{cooling}\right)$$
where $T_{i}$ is a medium temperature at the *i*th step, and $∆T_{heating}$ and $∆T_{cooling}$ are contributions to change in the medium temperature due to heating and cooling systems. Individual temperature change contributions $∆T_{heating}$ and $∆T_{cooling}$ are defined in Equations (4) and (5):(4)$$∆T_{heating}={heating}_{i-1}^{\prime }\cdot \frac{P_{heater}\cdot ∆t}{m_{water}\cdot c_{p,water}}$$(5)$$∆T_{cooling}={cooling}_{i-1}^{\prime }\cdot \frac{\dot{m}\cdot {cp}_{water}\cdot \left(T_{cold\;water}-T_{i-1}\right)\cdot ∆t}{m_{water}\cdot c_{p,water}}$$
where ${heating}_{i-1}^{\prime }$ and ${cooling}_{i-1}^{\prime }$ represent heating and cooling actions performed by the agent. These incorporate the delayed response effects, defined in Equation (6):(6)$${heating}_{i-1}^{\prime }=min\left(\frac{{count}_{heatingON,i}}{{delay}_{heating}},1\right)$$
where ${delay}_{heating}$ is the heating delay parameter (the unit is the number of steps) and ${count}_{heatingON}$ is variable counting how many times heating was in the switched-ON state. This is defined in Equation (7),(7)$${count}_{heatON,i}=\left\{\begin{array}{c} {count}_{heatON,i-1}+1;\;if\;{heat}_{i-1}==1\\ {count}_{heatON,i-1}-1;\;if\;{heat}_{i-1}==0 \end{array}\right.$$

For which the following applies:(8)$$0\leq {count}_{heatingON,i}\leq 2\cdot {delay}_{heating}$$

The delayed response of the cooling system is simulated in the same manner as the heating system’s delayed response, as defined in Equations (9)–(11):(9)$${cooling}_{i-1}^{\prime }=min\left(\frac{{count}_{coolingON,i}}{{delay}_{cooling}},1\right)$$(10)$${count}_{coolON,i}=\left\{\begin{array}{c} {count}_{coolON,i-1}+1;\;if\;{cool}_{i-1}==1\\ {count}_{coolON,i-1}-1;\;if\;{cool}_{i-1}==0 \end{array}\right.$$(11)$$0\leq {count}_{coolingON,i}\leq 2\cdot {delay}_{cooling}$$

Learning was performed in 10 million steps, as shown in Figure 5, which took about 4 h on the workstation with an Intel Core i9 processor.

In the learning phase, the agent must follow the adjusted target temperature profile (10 min at 90 ± 2 °C, 10 min at 60 ± 2 °C, and then 10 min at 57 ± 2 °C). The deviation from the target temperature profile lies in maintaining 90 °C for 10 min instead of just 1 min, as specified in the target application. This adjustment ensures that the agent does not undervalue performance at the reference temperature of 90 °C compared to lower target values of 60 °C and 57 °C. During learning, the policy was saved and evaluated every 50,000 steps. After learning was completed, the best-performing policy was selected from all saved checkpoints. The optimal policy was found at 9,850,000 steps. Figure 5 shows the episode reward during training, indicating that the RL agent achieved good performance in simulation despite considerable variability in several simulation parameters, Table 1.

The reward exceeded 96% of the maximum achieved reward during the last 10% of the training steps, corresponding to 200 cooking episodes (1,000,000 steps/5000 steps per episode = 200 episodes). These episodes represent simulated cooking processes with varying combinations of influential parameters, all of which achieved rewards within a ±2% margin. This outcome serves as a measure of robust process control.

### 2.5. Reinforcement Learning Algorithm Parameters and Workstation Configuration

A DQN algorithm was selected to learn the policy to control the apparatus. The learning process was implemented using Python 3.9.18 programming language and a programming RL library Stable-Baselines3 [37]. The used implementation is standard Deep Q-Learning. Default DQN parameters settings (Table 2), suggested by the RL library, performed well empirically and were used in the learning process.

The workstation was configured with Windows 10, Python 3.9.18, Stable-Baselines3 2.2.1, PyTorch 2.1.2 (CPU version, GPU disabled), NumPy 1.26.2, Cloudpickle 3.0.0, and Gymnasium 0.29.1. The DQN agent code is available in the Appendix A. Although the reinforcement learning framework applied in this study could in principle also be implemented using commercial tools such as the MATLAB Reinforcement Learning Toolbox R2025a (version 25.1), the Stable-Baselines3 library in Python was selected because it is open-source, widely adopted in the research community, and offers greater flexibility for designing customized reward functions.

### 2.6. The Apparatus Control

After finishing the training process, the trained model was exported and transferred to the real system where the performance was evaluated, as shown in Figure 6. The actuator control interfaces were implemented using Arduino Uno microcontroller development boards, combined with power switches. The temperature acquisition units and actuator control interfaces were connected to the workstation via USB ports, as shown in Figure 6. The heater and the electromagnetic valve are governed by the agent, the hot water pump ran continuously, and the cold-water pump and the fan were governed by a simple utility program. The temperature sensors were Pt100—platina, 100 Ω at 0 °C, IEC 60,751 class A. The sampling frequency of the control system was *f* = 1 Hz. The location of the hot water temperature sensor is presented in Figure 2 and Figure 13, where the sensor is on the right, and the other sensor was used for testing purposes with the prototype only.

Figure 7 shows the egg cooker without its external casing, the workstation running the control program in the Stable-Baselines3 Python environment, the temperature acquisition units, the control interface to the heater and to the water exchange valve (see Figure 2), and the actuator control interface for devices that do not influence the temperature profile (i.e., pumps and fan; see Figure 2).

Figure 8 shows the temperature profiles over 2400 s (i.e., 34 min) under minimum, nominal, and maximum egg cooker heat capacities, by the first row of Table 1. The process begins with a heating phase up to 90 °C, followed by a 1 min hold at 90 °C. This is followed by cooling to 60 °C, a 10 min hold at 60 °C, further cooling to 57 °C, and finally a holding phase at 57 °C.

The duration of the heating phase up to 90 °C is determined by the load heat capacity, and by heating power, which depends on the actual electrical voltage (230 V ± 15%) in a single-phase outlet with a 16 A fuse.

A detailed view of the temperature curve from time *t* = 1180 s to *t* = 1300 s (temperature holding at 60 °C) in Figure 8, the nominal heat capacity curve, is presented in Figure 9**.**

The upper part of Figure 9 shows temperature holding at 60 °C, while the lower part illustrates the corresponding control actions:
Heater off, valve closed;Heater on, valve closed;Heater off, valve open;Heater on, valve open;Heater off, valve open to 20%.

The first event marked in Figure 9 shows a brief opening of the valve in control action 5. After delay, the temperature subsequently drops by 0.17 °C. The sequence indicates that the DQN agent has internalized understanding of the delay between action and system response. The valve closes before the water temperature begins to decrease.

A short heating pulse is applied 27 s after the cooling pulse. Again, temperature increases with a time delay and the heater is turned off before the temperature begins to rise. This demonstrates that the agent considers the time lag between the action and thermal response.

The temperature, within the annotated period, remains within T ± 0.03 °C for a duration of 50 s, despite using relatively coarse temperature control, where the system operates in 1 s discrete time intervals, with heating power of 3.6 kW.

The temperature, within the annotated period, is approximately 0.04 °C below the nominal 60 °C setpoint. The reason is most likely in the reward structure, used during DQN training: the agent is rewarded not only for minimizing temperature error, but also for minimizing fluctuations in the slope of the controlled temperature. Once the temperature is sufficiently close to the reference temperature, the agent is additionally rewarded for maintaining it as constant as possible.

The temperature profiles at minimum and maximum load heat capacities, shown in Figure 8, are practically identical to the nominal load profile in the temperature-holding phases, despite the heated water volume ratio between the two cases being 1.0/2.4 (5 L/12 L).

The kink observed in the heating curve at the maximum heat capacity appears when the system transitions from continuous heating to regulated heating at 87 °C. Evidently, at constant power, heating larger water volume takes longer.

Further supporting the validity of our approach, Table 3 presents the mean squared errors (MSEs) of the RL controller. The MSEs are calculated for temperature curves from cooling to 60 °C further on. While the fuzzy controller [22] performs adequately under nominal laboratory conditions, prolonged field use has revealed limitations. Some users, in an attempt to avoid excessive firmness, reduced the holding time at 90 °C to a minimum value, despite the absence of any physical justification, which may compromise pasteurization reliability. In addition, after extended holding periods (up to 6 h at 57 °C), eggs were reported to become noticeably firmer than freshly processed ones. These findings highlight the reduced robustness of the fuzzy controller under prolonged field operation. In contrast, the RL-based controller, trained with parameter distribution densities, consistently achieves low MSEs across all simulated scenarios. In 300 simulation runs with parameter probability densities, the MSE ranged from 0.0271 to 0.0292. On the target system, the MSE was 0.0620 under minimum load and 0.0604 under maximum load, as shown by the curves in Figure 8. Verification of the RL-based controller on the target system will require additional work, as discussed in Section 2.8.

### 2.7. The Apparatus Mechanical Improvements

Heat redistribution processes take time, either by conduction, convection, or radiation. In our case, forced convection dominates over conduction, and radiation is negligible.

The heater and the system temperature sensor are positioned at a short distance from each other, which is a necessity for effective temperature regulation. Of particular interest and importance is the delay between temperature changes at the heater and at the water-immersed eggs. Figure 10, left, displays the temperature curve measured at the center of the hot water container, at full heater power, from the start of the process up to 90 °C. Figure 8, right, illustrates the repeatability of the temperature acquisition unit; in this case, the measured temperature is T = 21.76 ± 0.01 °C.

The temperature curve in Figure 10 left exhibits significantly greater fluctuation compared to the curves in Figure 8, despite the heater being continuously on in both cases.

The key finding from the analysis of Figure 10 is the issue of ineffective convective heat transfer. The pump-assisted movement of the heated water does not generate laminar circulation throughout the heated volume. As a result, even a carefully designed temperature regulation system becomes of a questionable value—except at the reference point near the heater, where the system temperature sensor is located and where temperature control performance is benchmarked.

After consulting with experts in thermal system design for pharmaceutical and food-processing equipment [39], we performed a redesign of the egg cooker’s hardware details. The goal was to achieve uniform, laminar circulation of water throughout the heated vessel.

The modifications implemented, in Figure 11, compared to the apparatus in Figure 2—were the following:Water is now introduced into the vessel through a top-side inlet, while maintaining upward circulation from the bottom, in order to enhance overall water distribution.The three-way valve (Figure 2) was replaced with a simpler on/off valve (Figure 11), improving water circulation during the cooling phase. A secondary benefit is that the cooling power now exceeds the heating power to a lesser extent than in the original design, bringing the system closer to thermal symmetry.The thermal insulation of the inner vessel was improved. The new insulation is dual-layered. The inner insulation layer resists condensation, which can degrade conventional fibrous insulation materials. This protects the outer high-performance layer from moisture exposure and helps maintain long-term thermal performance. Assuming a thermal conductivity of approximately 0.020 W/m·K for aerogel and 0.08 W/m·K for silicone foam, the combined *R*-value of a 10 mm wall can reach approximately 0.625 (K·m^2^/W), significantly outperforming single-layer alternatives of the same thickness.

Figure 12 shows thermal images of the egg cooker after the modifications. The temperature field within the water is practically uniform and clearly separated from the rest of the apparatus.

Figure 13 displays the positions of the temperature sensors in the casing, water, and egg, used to evaluate the thermal distribution after the egg cooker’s improvements. Figure 13, left: The sensor on the right is used for temperature regulation, and the other one is used for temperature monitoring in the prototype. Figure 13, middle and right: A custom holder was made to accommodate a water temperature probe and up to two eggs to measure their internal temperature using thermocouple probes.

Figure 14 shows a somewhat impractical method for measuring mid-yolk temperature in a water-submerged egg. First, a hole is drilled through the shell, a thermocouple temperature probe is inserted into the center of the egg, and the cable entry point is temporarily sealed with a cyanoacrylate adhesive.

Figure 15 shows three temperature profiles during the egg-cooking process: the temperature at the regulation sensor, the water temperature, and the mid-yolk temperature inside the egg. The pasteurization temperature and time are highlighted in Figure 16.

In Figure 15, the temperature curve measured at the regulation sensor is about the same as that before the hardware modifications. The water temperature is shown as a smooth blue curve. The mid-yolk temperature curve follows the other two with a noticeable delay.

Mechanical modifications to the apparatus improved the thermal distribution to the extent that the circulating water now functions as a thermal filter—that is, it effectively filters out the small oscillations resulting from temperature regulation at the heater. The mid-yolk temperature, measured with a thermocouple at a resolution of 0.1 °C, shows no detectable regulation induced temperature oscillations.

In Figure 15 and Figure 16, the initial egg temperature is 3 °C (1 °C lower than the 4 °C industrial standard). The blue rectangles mark the time–temperature region that satisfies pasteurization criteria: a temperature of T ≥ 60 °C maintained for *t* ≥ 10 min. In our case, the pasteurization time *t*_p_ = 12.25 min, which is adequate, based on our internal studies [24,25,26], and is consistent with the recent findings in [40,41].

Simultaneously, the egg exhibits the expected organoleptic and visual properties following pasteurization, as shown in Figure 14, right.

The arrows in Figure 15 illustrate the mechanism for fine-tuning the organoleptic properties of pasteurized soft-boiled eggs. Small incremental increases in holding time at 90 °C (in 10 s steps, since yolk coagulates already at 65 °C [42]) accordingly raise the egg’s internal temperatures, resulting in a firmer texture. This feature is intended to support the market differentiation of the apparatus, allowing chefs to adjust egg texture based on personal or culinary preferences.

### 2.8. Methods for Porting the Algorithm to an Embedded System

We developed a simulator of the Egg cooker’s thermal behavior, formulated a reward structure for reinforcement learning, and used both to train a DQN agent to control the thermal process for cooking pasteurized soft-boiled eggs. The performance of the trained DQN agent was evaluated using the physical egg cooker. During evaluation, we improved the operation of the egg cooker itself.

The DQN agent successfully controls the thermal process consistently across both nominal and extreme operating conditions. This demonstrates the robust control of a thermally sensitive process—constituting the application-level achievement. The methodology of virtual training for environment-robust foods processing, combined with verified deployment on real hardware represents the primary scientific and developmental contribution of this work.

For commercial deployment, the industrialization of DQN-based thermal process control is required, for which several implementation pathways are available. The most influential factors in selecting an approach are the apparatus production volume, target production cost, and permitted development time.

#### 2.8.1. Option A: Laboratory Python-Based DQN Control

The thermal process is controlled using the trained DQN agent running in a Python environment with PyTorch 2.1.2 and Stable-Baseline3 2.2.1 libraries. This setup functions reliably, and it is perfectly suitable for research and prototyping in laboratory. It cannot be a reasonable choice for a commercial industrial application. The hardware requirements are cost-prohibitive; the system setup and the user interface are too complex for product deployment.

#### 2.8.2. Option B: Deployment via ONNX and NVIDIA Jetson

The trained DQN agent is exported from Python into the ONNX (Open Neural Network Exchange) format, preserving the model structure and weights. It is deployed on an NVIDIA Jetson microcontroller with a CUDA-(Compute Unified Device Architecture, for running AI algorithms) capable GPU, which is priced with the acceptable range for embedded system design (under EUR 200). This hardware supports both ONNX Runtime and TensorRT, making it well-suited for neural network-based applications.

This option allows for low-volume production using developer kits. For mid-volume production (<50 units), a custom PCB (Printed Circuit Board) would be needed for interfacing, control, and custom GUI (Graphical User Interface).

#### 2.8.3. Option C: TensorFlow Lite (TFLite) Deployment

The DQN neural network is implemented using Google’s TensorFlow Lite (TFLite) framework. The neural network is implemented via optimized software libraries. TFLite Micro enables running neural networks (e.g., ReLU, SoftMax) on lightweight platforms using C++. Supported platforms include Raspberry Pi and ARM Cortex-A, making this solution cost-effective for small or mid-sized neural networks like the one used in this study.

#### 2.8.4. Option D: Apache TVM for Bare-Metal Deployment

Apache TVM v2.4.65 (Tensor Virtual Machine), developed by the Apache Software Foundation, is an open-source framework that compiles a smaller neural network into optimized C code. TVM supports deployment on resource-constrained microcontrollers, including those without an operating system.

#### 2.8.5. Option E: Manual Optimization and Target-Specific Coding

Manual conversion into target-specific code, requiring strong determination and a significant investment of time and some effort.

For the deployment of the thermal process control suited DQN agent in an embedded system, options D and C are the most suitable. The preferred approach is a bare-metal embedded system without an operating system, as it allows for instant startup/shutdown, and simplified real-time operation with no need for a real-time operating system (RTOS). In the best case—depending on the neural network’s complexity—the exported and C-coded neural network is self-sufficient, i.e., it does not need any library to support its implementation. However, it is still a black box, regarding reading functionality from its weights and activation functions.

In regulated sectors such as food and pharmaceuticals, interpretability and verifiability are crucial components of any design. They are built into the companies’ policy at minimum in the form of periodical design reviews by seniors and project leaders. Logic based on conditional statements (*if–else* rules) enables transparent verification, certification, clear understanding of decision-making processes, and allows for manual adjustments.

A C-coded (C is currently a de facto standard for embedded system coding) neural network presents a white-box code of black-box behavior, for which functionality can be verified via extensive testing and testing on reference cases, that cover all possible uses. This approach, supported with new verification developments, will have to evolve into a standard for the verification of neural networks functionality.

## 3. Discussion

Unlike conventional control programs written in C-style logic, where behavior is explicitly coded via conditional statements, deep neural networks use weighted, non-linear transformations to implicitly encode decision rules. This allows learned controllers to generalize over diverse environmental conditions and achieve robust performance without requiring the manual enumeration of all operational cases. The key theoretical foundation behind this capability is the Universal Approximation Theorem [43,44,45]. It states that a feedforward neural network with a single hidden layer containing a finite number of neurons, and using a non-linear activation function (such as ReLU or sigmoid) can approximate any continuous function on a compact subset of Rⁿ, to any desired degree of accuracy. This means that a neural network can learn and perform mappings as complex as those defined by arbitrarily nested *if–else* logic, provided sufficient data and training are available. This allows DQN-based controllers to generalize over diverse environmental conditions and execute complex control strategies that would be difficult to encode manually using *if–else* logic.

In our case, the DQN agent learned to control the thermal process for cooking soft-boiled eggs under a range of starting and environmental conditions. The agent consists of a fully connected neural network with two hidden layers of 64 ReLU neurons each. This architecture learns a non-linear function that maps input observations (recent temperature history) to optimal control actions (heating and cooling the in-water immersed eggs). Each ReLU activation introduces a segment-wise linearity that enables the network to effectively partition the input space into regions corresponding to different optimal actions—akin to implicit *if–else* rules. The network, through training, generalizes across different thermal conditions and achieves consistent cooking outcomes. This behavior is characteristic of robust reinforcement learning, in which the agent learns to adapt its policy over a range of operational variabilities and disturbances, while maintaining performance. The DQN policy is continuous and generalizable, and was learned directly from model data via robust reinforcement learning. This property is a hallmark of robust reinforcement learning [46], in which the agent develops a policy that performs reliably even under varying and uncertain operating conditions.

Although rule-based control logic using *if–else* statements remains a common and interpretable method in industrial process control, its practicality significantly diminishes as system complexity increases. For thermally sensitive, non-linear, and time-dependent processes—such as pasteurized soft-boiled egg cooking under variable initial and environmental conditions—the number of required rules grows rapidly. This leads to rule sets that are difficult to manage, maintain, and optimize, especially when interaction effects (e.g., heat transfer dynamics, delays) must be considered.

In contrast, our DQN agent, trained via robust reinforcement learning, learns policy that handles these variations smoothly and adaptively, without the need for explicitly encoding all conditional branches. The agent’s neural network generalizes well across the observed state space and accounts for time delays, thermal inertia, and cross-variable dependencies.

Additionally, while some tools exist for automatically extracting symbolic rules—decision trees or program synthesis from policy trajectories [47,48], they are still limited in their abilities, compared to deep neural policies.

Therefore, although symbolic logic remains most valuable for verification or auditing, our results indicate that a robustly trained neural policy offers a more scalable and maintainable solution for dynamic thermal control in the relatively delicate cooking process of making pasteurized soft-boiled eggs.

In our project, the use of a simulated environment was crucial for the training phase of the RL agent. Given the impracticality of long episodes in the real-cooking learning, the simulation provided a controlled setting where the agent performs millions of steps to achieve the desired performance. The domain randomization technique ensured that the simulation model closely mirrored real-world variability, enhancing the robustness of the RL agent in control of the thermal process in the actual apparatus.

## 4. Conclusions

We introduced reinforcement learning algorithms, particularly the DQN agent, for robust and precise temperature control. The developed system successfully addresses the challenges of maintaining consistent temperature, thereby ensuring both food safety and quality. The DQN agent demonstrated the ability to learn and adapt to various thermal conditions, overcoming the limitations of a fuzzy-logic system in handling different heat capacities, environmental temperatures, initial cooling-water temperatures, supply voltage variations, and performance degradation caused by limescale buildup on the heater surface. Our results indicate that RL-based temperature control enhances the robustness and efficiency of the thermal process, paving the way for further innovations in industrial food processing.

It should be noted that in this study, the agent was not tested under conditions outside the training distribution (Table 1). A functional egg cooker does not operate beyond these ranges; in fact, the apparatus design provides less freedom than was considered during agent training, with the difference representing an operational safety margin. Nevertheless, hardware failures, which may occur over the lifetime of any apparatus, could lead to conditions outside the training ranges. The industrialization phase of this project will therefore require testing the agent beyond the training ranges and developing measures to ensure unconditional operational safety.

Future work will focus on refining RL models and extending their applicability to other thermal processes in the food industry.

## Figures and Tables

**Figure 1 foods-14-03171-f001:**
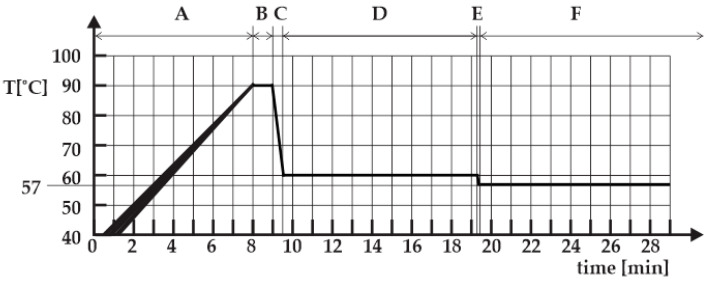
The reference temperature curve [23,24].

**Figure 2 foods-14-03171-f002:**
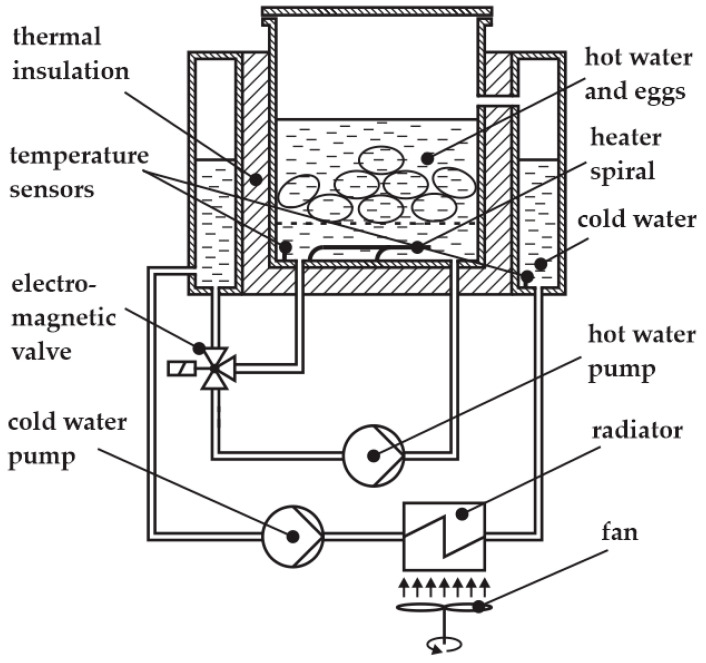
Schematic of the egg cooker.

**Figure 3 foods-14-03171-f003:**
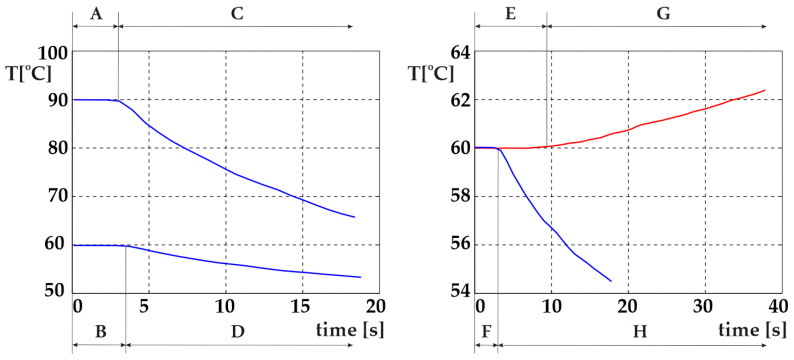
(**Left**): cooling from different temperatures, (**right**): different time delays of heating and cooling [22].

**Figure 4 foods-14-03171-f004:**
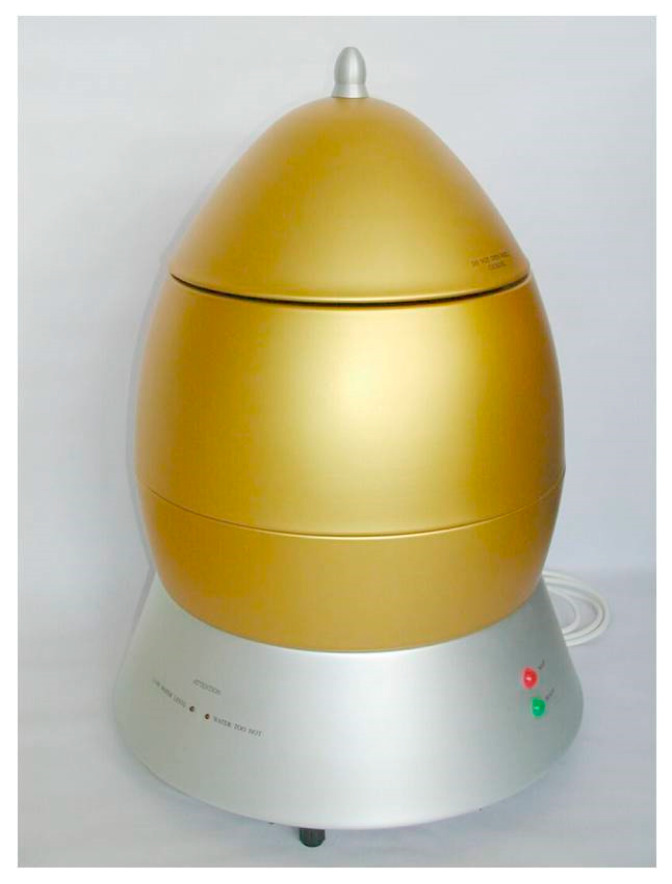
The egg cooker.

**Figure 5 foods-14-03171-f005:**
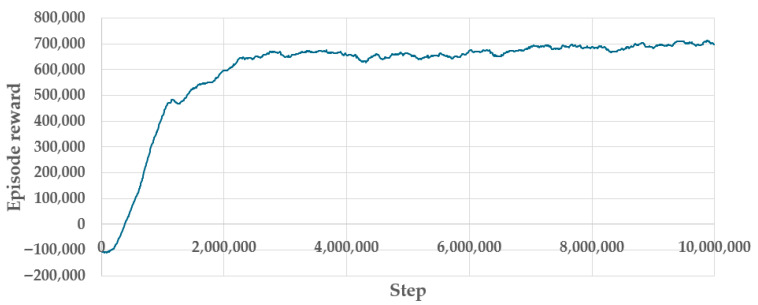
Episode reward during learning.

**Figure 6 foods-14-03171-f006:**
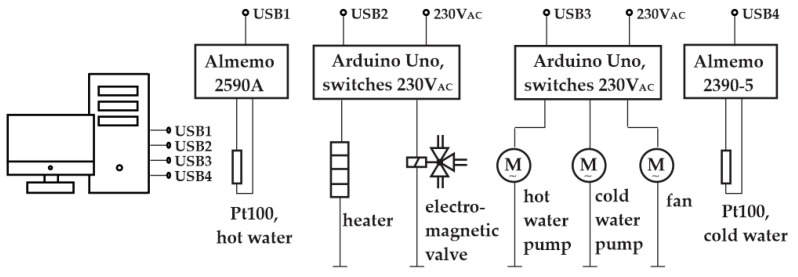
Setup for evaluation of the DQN controlled thermal process.

**Figure 7 foods-14-03171-f007:**
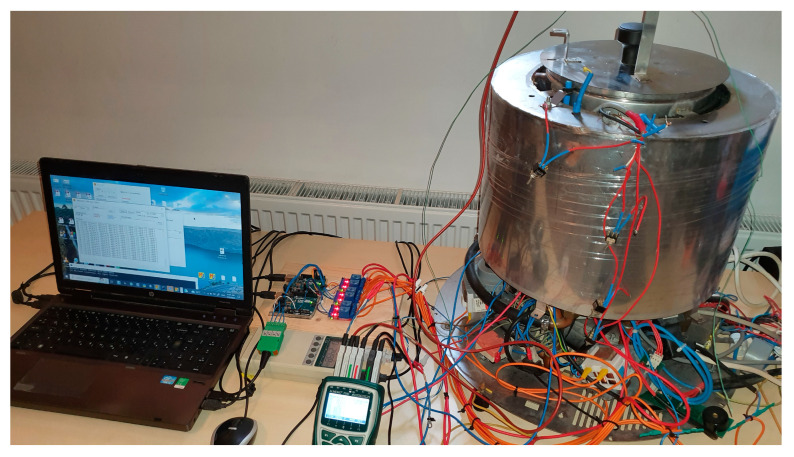
Physical setup for evaluation of the DQN controlled thermal process.

**Figure 8 foods-14-03171-f008:**
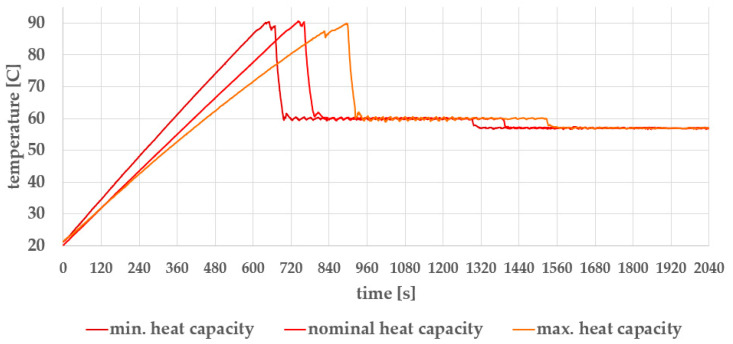
Temperature profiles at minimum, nominal, and maximum load heat capacity.

**Figure 9 foods-14-03171-f009:**
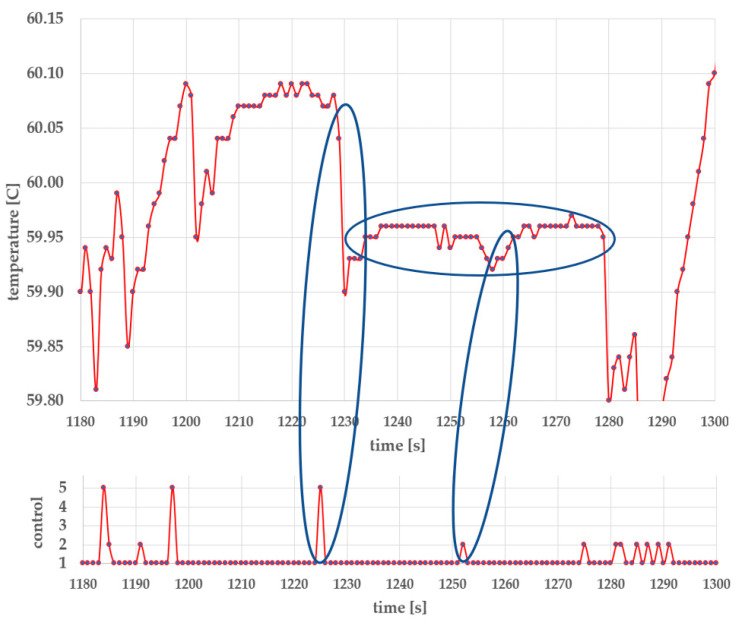
AI-based temperature holding control at nominal-load heat capacity.

**Figure 10 foods-14-03171-f010:**
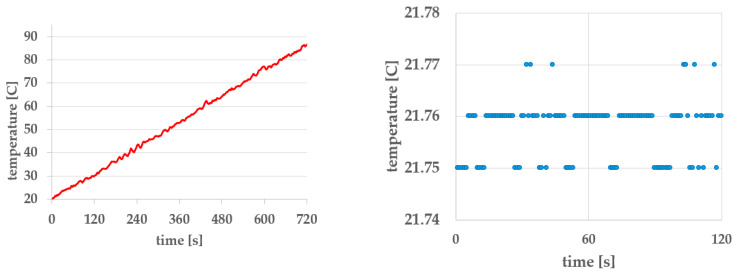
(**Left**): temperature curve measured at the center of the hot water container; (**right**): measurement repeatability of the temperature acquisition unit.

**Figure 11 foods-14-03171-f011:**
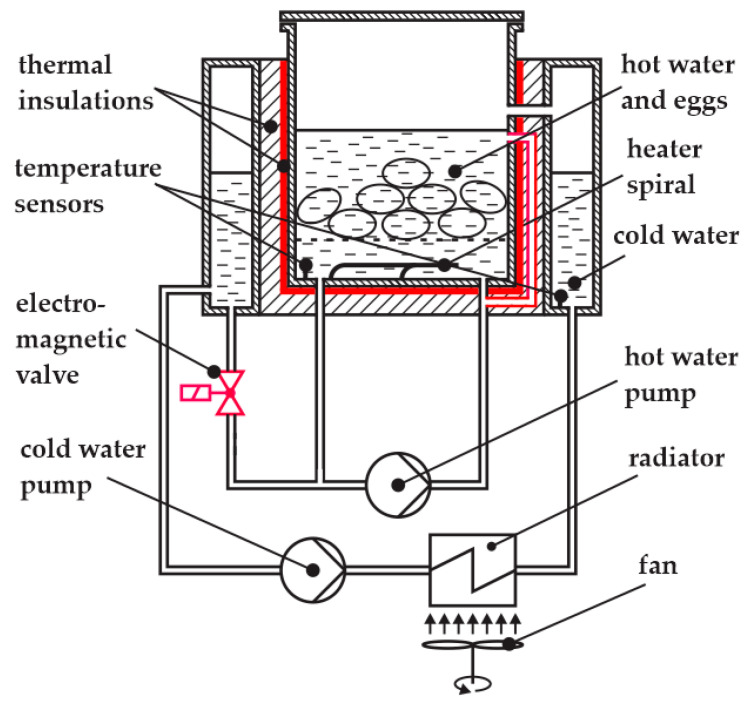
Thermally improved egg cooker.

**Figure 12 foods-14-03171-f012:**
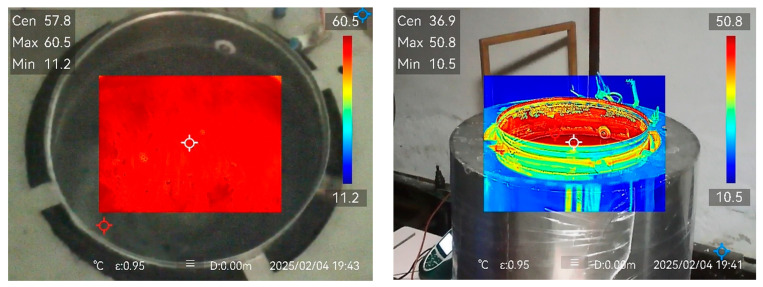
Thermal views of the improved Egg cooker.

**Figure 13 foods-14-03171-f013:**
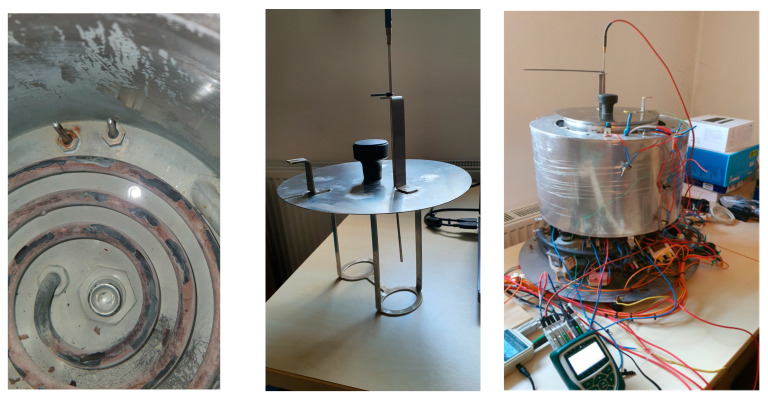
Positions of temperature sensors in casing, water and egg.

**Figure 14 foods-14-03171-f014:**
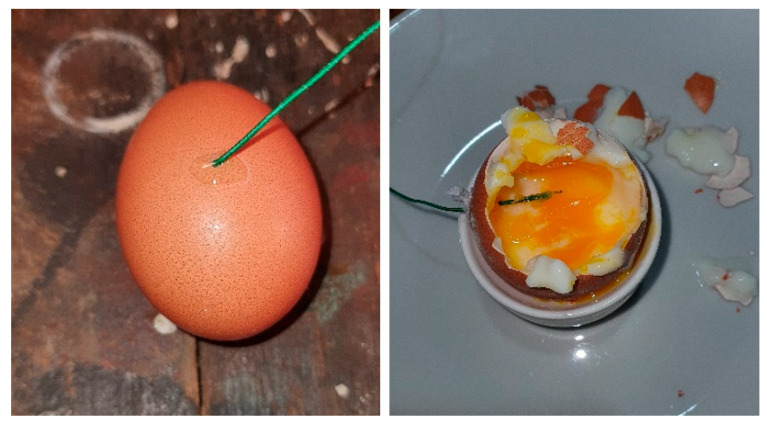
Setup for measuring mid-yolk temperature.

**Figure 15 foods-14-03171-f015:**
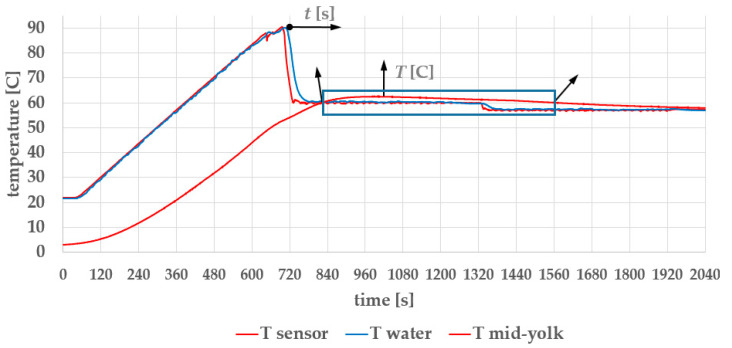
The three measured temperatures: at the system sensor, in the water, and in the mid-yolk.

**Figure 16 foods-14-03171-f016:**
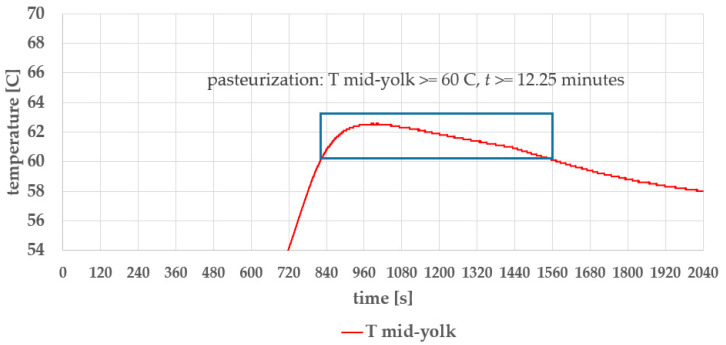
Pasteurization temperature and time.

**Table 1 foods-14-03171-t001:** Limits of randomly set simulation model parameters at the beginning of each episode, and nominal parameters values.

Parameter	Min. Value	Nominal Value	Max. Value
$m_{heated\;water}\;$ [kg]	5.00	7.50	12.00
$P_{heater}$ [W]	2300	3600	4800
$T_{cooling\;water}$ [°C]	10	20	35
$\dot{m}$ [kg/s]	0.1	0.2	0.3
${delay}_{heating}$ [s]	1	2.5	4
${delay}_{cooling}$ [s]	1	2	3
$\sigma_{T\;measurement\;noise}\;$	0.01	0.03	0.05

**Table 2 foods-14-03171-t002:** The Stable-Baselines3 DQN algorithm learning parameters.

Parameter	Value
Learning rate	0.0001
Buffer size	1 million
Learning starts	100
Batch size	32
*τ*	1.0
*γ*	0.99
Train frequency	4
Gradient steps	1
Target update interval	10,000
Exploration fraction	0.1
Exploration initial *ε*	1.0
Exploration final *ε*	0.05
Maximum value for the gradient clipping	10

**Table 3 foods-14-03171-t003:** Comparison of RL (DQN agent) performance against the reference temperature profile in simulation and in the target system, prior to apparatus improvements.

Controller	MSE [°C^2^]	Qualitative Outcome
RL (DQN agent) in simulation, N_RUNs_ = 300	0.0271–0.0292	Robust across scenarios, consistent cooking
RL (DQN agent) in target system	0.0604–0.0620	Verification: according to Section 2.8—Porting to an embedded system

## Data Availability

The raw data supporting the conclusions of this article will be made available by the authors on request.

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
