# Peer review of "AI Control for Pasteurized Soft-Boiled Eggs"

_foods, 2025, doi:10.3390/foods14183171_

Round 1

Reviewer 1 Report

Comments and Suggestions for Authors

This is a strong and promising manuscript that demonstrates the successful application of reinforcement learning to a real-world, high-precision thermal process in food preparation. The technical work is robust, and the integration of simulation, control theory, and hardware implementation is commendable. However, the manuscript would benefit from a more formal scientific tone, clearer framing of the contribution, and additional experimental validation.

Comments:

    • The introduction, while well-grounded in prior work, could benefit from a clearer delineation of the scientific contribution. At present, it occasionally reads as a product development report. I suggest framing the novelty more explicitly: e.g., what new knowledge does this study generate in the broader field of AI-based process control?
    • Certain sections, particularly those describing the etymology of reinforcement learning (RL) and references to personal anecdotes (e.g., dog training), are too informal and distract from the technical focus of the paper. These should be either significantly shortened or removed. The same point applies to general historical context on egg consumption—while potentially interesting, it does not directly support the research question.
    • The reward function is complex and appears to have been tuned empirically. A deeper discussion or ablation study on how the structure of the reward function affects agent performance would strengthen the credibility and reproducibility of the approach.
    • The manuscript states that training was done exclusively in simulation and then transferred to the physical system. However, there is little quantitative comparison between the simulated and real-world performance beyond qualitative plots. Consider including metrics (e.g., MSE between target and measured temperature profiles) to better support the validity of the sim-to-real transfer.
    • The RL agent’s robustness to unseen conditions is a central claim. However, only the nominal and boundary cases are shown. More discussion is needed on whether the agent was tested on conditions outside the training distribution, and how it performed.
    • The mechanical redesign of the egg cooker and its thermal insulation is a valuable part of the contribution. However, these improvements could be better separated from the core RL contributions, perhaps in an appendix or a dedicated subsection, to avoid blurring the focus.
    • Some figures (e.g., Figures 3, 7, 9, 13) lack proper captions and scale bars. Ensure all visual materials are self-explanatory and readable when printed in grayscale.
    • The term “DQN agent” is used throughout. Clarify whether this refers strictly to the standard DQN algorithm or a modified version (e.g., with target networks, prioritized replay, etc.).
    • The implementation relies on Stable-Baselines3 and other known packages. Consider providing a minimal configuration file or GitHub link for reproducibility (even as supplementary material).

Reviewer 2 Report

Comments and Suggestions for Authors

1. The necessity of using reinforcement learning for temperature control in soft-boiled egg preparation needs further justification. It appears that similar outcomes could be achieved using traditional feedback control or predefined control strategies. 

2. The experimental design requires improvement by adding a control group for comparison. 

3. The manuscript needs to better highlight the contribution. The novelty of the currenr version is not sufficiently evident compared with existing temperature control techniques. 

Reviewer 3 Report

Comments and Suggestions for Authors

The following answer is only targeted at the above-mentioned file, and any discussions about other files will be ignored. This paper proposes a reinforcement learning (RL)-based temperature control system for pasteurized soft-boiled eggs, aiming to address the deficiencies of traditional control methods under load changes and environmental disturbances. However, there are still some issues that need to be corrected. The specific problems are as follows:

1. Lines 1-7:The applications of AI in food processing mentioned in the introduction (lines 1-7) are mostly review literature and lack core research citations from the past three years. It is recommended to supplement the latest relevant technical papers to enhance the timeliness and relevance of the literature.

2. Lines 48-59: The introduction spends a significant amount of space (lines 48-59) discussing the history, consumption, nutritional value, and health controversies of eggs, which are less relevant to the core technical content of the paper. It is recommended to simplify this section and focus on the technical challenges of AI in food processing, particularly the shortcomings of existing control methods, to highlight the research motivation.

3. Lines 125-134: When introducing the origins of RL, the author spends a lot of words describing the connection between personal dog-raising experience and RL (lines 125-134), which deviates from academic rigor. It is suggested to delete such personal experiences and replace them with a more direct technical background explanation to enhance the academic nature of the paper.

4. Lines 133-180:Although the description of RL components is detailed, it does not explain the construction details of the simulation environment (such as physical models, parameter settings), the source of the training dataset, and key information such as hyper parameter configuration (such as learning rate, batch size), resulting in insufficient reproducibility of the method.

5. Lines 75-81: In the description of the device in Figure 2, only the heating and cooling mechanisms are briefly explained, but the type and location of sensors, the sampling frequency of the control system, and the switching logic between PID and RL controllers are not specified. It is recommended to supplement the detailed architecture diagram of the hardware configuration and control.

6. Lines 103-118: The issues mentioned by the author, such as load sensitivity and batch variation, are based on observations from actual use, but have not been validated in experiments by simulating these scenarios. It is recommended to design experiments to simulate different loads, continuous batches, and environmental temperature changes to verify the robustness of RL.

7. Lines 76-81, 100-101: Figure 1 does not indicate the temperature range and key time nodes, and Figure 3 lacks specific temperature values and explanations of experimental conditions. It is recommended to improve chart annotations and supplement explanations of data sources.

Round 2

Reviewer 1 Report

Comments and Suggestions for Authors

The revised version of your manuscript shows substantial improvement compared to the original submission. I would like to acknowledge the effort you invested in carefully addressing the comments.

One limitation that remains is that the RL agent was not tested under conditions outside the training distribution. While you have explained this clearly and justified postponing such tests to the industrialization phase, it would be helpful to emphasize this limitation in the Conclusions section as an explicit avenue for future work.

Overall, I am satisfied with the revisions. The manuscript has been substantially improved and now convincingly demonstrates the contribution of reinforcement learning to precise thermal process control in food applications.

Reviewer 2 Report

Comments and Suggestions for Authors

It can be accepted.
